# Impact of variants of concern on SARS-CoV-2 viral dynamics in non-human primates

**Aurélien Marc**[1]*, **Romain Marlin**[2], **Flora Donati**[3,4], **Mélanie Prague**[5,6], **Marion Kerioui**[1], **Cécile Hérate**[2], **Marie Alexandre**[5,6], **Nathalie Dereuddre-bosquet**[2], **Julie Bertrand**[1], **Vanessa Contreras**[2], **Sylvie Behillil**[3,4], **Pauline Maisonnasse**[2], **Sylvie Van Der Werf**[3,4], **Roger Le Grand**[2], **Jérémie Guedj**[1]

**1** Université Paris Cité, IAME, Inserm, Paris, France, **2** Université Paris-Saclay, Inserm, CEA, Center for Immunology of Viral, Auto-immune, Hematological and Bacterial diseases (IMVA-HB/IDMIT), Fontenay-aux-Roses and Le Kremlin-Bicêtre, Paris, France, **3** National Reference Center for Respiratory Viruses, Institut Pasteur, Paris, France, **4** Molecular Genetics of RNA Viruses Unit, Institut Pasteur, UMR3569, CNRS, Université de Paris, Paris, France, **5** Inria Bordeaux Sud-Ouest, Inserm, Bordeaux Population Health Research Center, SISTM Team, UMR 1219, University of Bordeaux, Bordeaux, France, **6** Vaccine Research Institute, Créteil, France

* aurelien.marc@inserm.fr

**Data Availability Statement:** All animal data, statistical analysis code and modelling files are available on Zenodo at link https://doi.org/10.5281/zenodo.7304183.

## Abstract

The impact of variants of concern (VoC) on SARS-CoV-2 viral dynamics remains poorly understood and essentially relies on observational studies subject to various sorts of biases. In contrast, experimental models of infection constitute a powerful model to perform controlled comparisons of the viral dynamics observed with VoC and better quantify how VoC escape from the immune response. Here we used molecular and infectious viral load of 78 cynomolgus macaques to characterize in detail the effects of VoC on viral dynamics. We first developed a mathematical model that recapitulate the observed dynamics, and we found that the best model describing the data assumed a rapid antigen-dependent stimulation of the immune response leading to a rapid reduction of viral infectivity. When compared with the historical variant, all VoC except beta were associated with an escape from this immune response, and this effect was particularly sensitive for delta and omicron variant ($p < 10^{-6}$ for both). Interestingly, delta variant was associated with a 1.8-fold increased viral production rate ($p = 0.046$), while conversely omicron variant was associated with a 14-fold reduction in viral production rate ($p < 10^{-6}$). During a natural infection, our models predict that delta variant is associated with a higher peak viral RNA than omicron variant (7.6 $\log_{10}$ copies/mL 95% CI 6.8–8 for delta; 5.6 $\log_{10}$ copies/mL 95% CI 4.8–6.3 for omicron) while having similar peak infectious titers (3.7 $\log_{10}$ PFU/mL 95% CI 2.4–4.6 for delta; 2.8 $\log_{10}$ PFU/mL 95% CI 1.9–3.8 for omicron). These results provide a detailed picture of the effects of VoC on total and infectious viral load and may help understand some differences observed in the patterns of viral transmission of these viruses.

## Author summary

SARS-CoV-2 has been characterized by the successive emergence of Variants of Concern (VoC) that have caused large epidemic rebounds. However, as VoC emerged in very

**Funding:** This work was funded by the Bill & Melinda Gates Foundation through INV-017335 (JG). The NHP experiment was part of BIOVAR and PRODEVA programs funded by the ANRS-MIE project EMERGEN (ANRS0151). The funders had no role in study design, data collection and analysis, decision to publish, or preparation of the manuscript.

**Competing interests:** The authors have declared that no competing interests exist.

different contexts of pre-existing immunity, the comparison of their intrinsic effect of viral dynamics and infectiousness remains poorly understood. Here we analysed data from 78 non-human primates infected by different VoC (Beta, Delta, Gamma and Omicron BA.1) and we used a mathematical model to quantity the impact of VoCs on viral load and infectivity. Compared with the historical variant, Omicron and Delta variants were associated with a longer and larger excretion of infectious particles, which was attributed in the model to an enhanced capability to escape the immune response. While Delta variant was associated with a larger peak viral load than the historical variant, no such effect was observed for Omicron variant. This suggests that the increased transmissibility of Omicron variant does not stem from higher viral load levels but rather from its ability to maintain high levels of infectious particles over time. Altogether, these results illustrate the importance of quantifying both viral load and infectiousness to better understand some differences observed in the patterns of viral transmission of VoCs.

## Introduction

The sever acute respiratory coronavirus 2 (SARS-CoV-2) is the causative agent of the Coronavirus-induced disease 2019 (COVID-19) cumulating more than 500 million cases and over 18 million death as measured by excess mortality as the end of 2022 [1,2]. Repeatedly, several variants have emerged and although most of them vanished quickly, some of them, called Variants of Concern (VoC), in particular alpha, beta, gamma, delta and omicron have caused dramatic epidemic rebounds [3–5]. These variants have acquired specific mutations enhancing their infectious capacities and escaping the immune response, leading to a dramatic loss of efficacy of monoclonal antibodies [6]. They have also caused a large drop in vaccine efficacy against disease acquisition even though until now vaccine remain largely effective against severe disease [7–9].

While several millions of individuals have been infected by these VoC, we still do not have a precise understanding on the effects of VoC on viral load. Even though some effects on larger levels of viral excretion have been reported [10–13], these studies often lack of robustness, and may be biased by many confounding factors that complicate comparisons, in particular reporting biases, heterogeneity in the incubation period and vaccination coverage.

In that context where human clinical data are difficult to interpret, the non-human primate (NHP) experimental model offers a unique opportunity to describe infection with SARS-CoV-2 in detail in a fully controlled environment. Since 2020, our group has conducted many studies to evaluate the effects of antiviral drugs or vaccines in this model [14,15], and showed its large predictive value [16]. Here, we analysed retrospectively viral load data obtained in 78 animals that were included as control arms of these studies and that were infected with different strains of SARS-CoV-2 (historical, beta, gamma, delta and omicron (BA.1)). In addition, we performed longitudinal measures of viral culture to evaluate a potential effect of VoC on viral infectivity. Using the techniques of mathematical modelling, we characterize the viral kinetics in these animals and we discuss their biological insights.

## Results

### Variant of concern viral kinetics

Several biomarkers were measured, both genomic RNA and subgenomic RNA were quantified at regular interval over all the study period and infectious titers at 2 times points. All macaques developed a rapid infection with genomic viral load peaking between 2- and 3-day post-infection (dpi) for the historical and beta variant, 3.5 dpi for variant delta and 4 dpi for variants

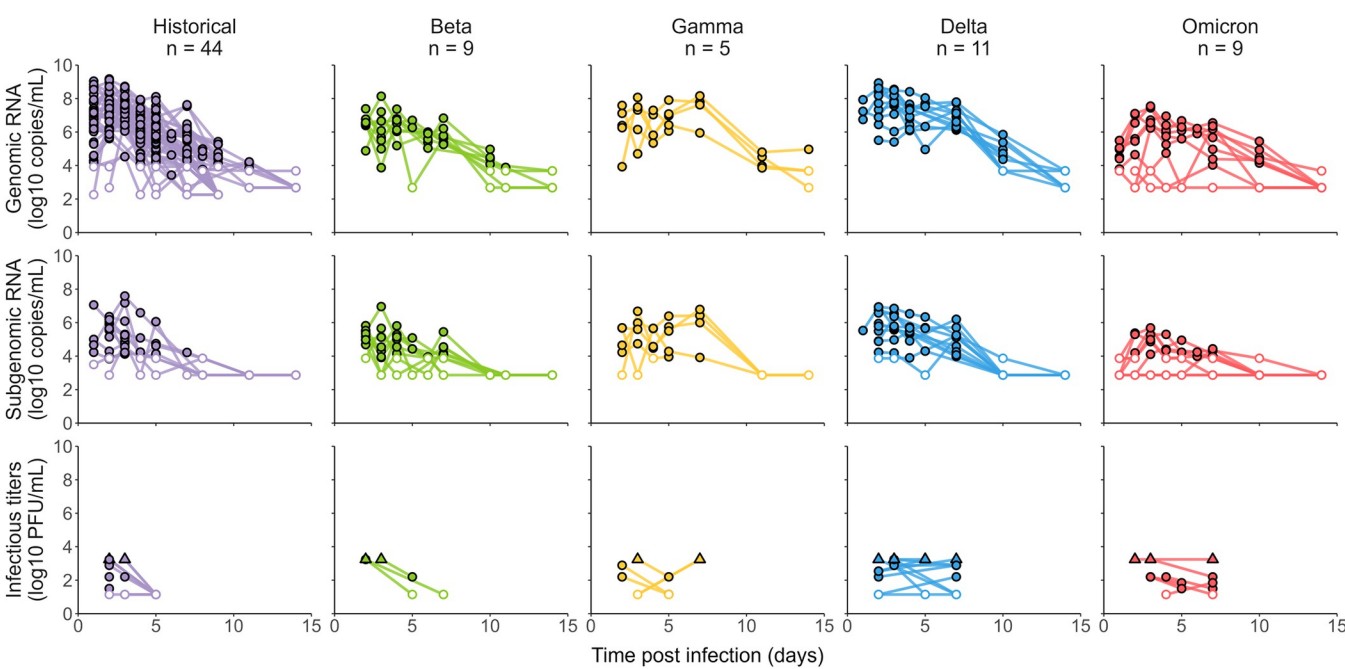

**Fig 1. Longitudinal measurements of genomic RNA, subgenomic RNA and infectious titers in 78 infected cynomolgus macaques.** Both limit of quantification and detection are depicted as empty dots, the latter being lower. Upper limit of detection is depicted as filled squares.

gamma and omicron (BA.1). Genomic viral load was cleared at 8 dpi for the historical variant, 10 dpi for the beta variant, at 12 dpi for variants delta and omicron (BA.1) and at 14 dpi for variant gamma (Fig 1 and S1 Table). In addition to viral RNA, infectious titers were measured for 41 animals. Infectious titers were measured by Tissue Culture Infectious Dose (TCID$_{50}$) from nasopharyngeal swab sampled at 2 time points per animal (day 2, 3 or 4 plus at day 5 or 7 post-infection). As we included several control animals from different studies, infected with either TCID$_{50}$ or Plaque Forming Units (PFU), all TCID$_{50}$ were converted to PFU assuming 1 PFU = 0.7 TCID$_{50}$ [17]. All infectious titers quickly dropped to undetectable levels for the historical variant at 5 dpi, where for the other variants the infectious titers remained consistent over the course of the infection (Fig 1).

## Viral dynamic model

To account for the quick drop in infectious titers observed in the historical variant, (Figs 1 and S1) several models incorporating an action of an antigen-mediated immune response were tested (Fig 2). All models, except a model targeting the viral production parameter, provided an improvement of BIC compared to a target cell limited model (Table 1). We found that a model targeting the infectious ratio best described our data (Model 1 in Table 1). In the following, we discuss the parameter values of the final constructed model accounting for both an effect of the immune effector and variant specific effect on the parameters (see below). For the historical variant, we estimated the infectivity rate parameter $\beta$ at $1.86 \times 10^{-5}$ copies$^{-1}$.d$^{-1}$ (95% confidence interval (CI) $1 \times 10^{-5}$–$3.39 \times 10^{-5}$) and the loss rate of infected cells $\delta$ at 1.38 d$^{-1}$ (95% CI 1.22–1.55), corresponding to a half-life of 12 hours. We estimated the viral load production parameter $p$ at $9.44 \times 10^{5}$ copies.cells$^{-1}$.day$^{-1}$ (95% CI $2.1 \times 10^{5}$–$1.68 \times 10^{6}$). This corresponds to a within-host basic reproductive number $R_0$ (i.e., the number of newly infected cells by one infected cell at the beginning of the infection) of 3.1 (95% CI 2–4.3) and a burst size (i.e the

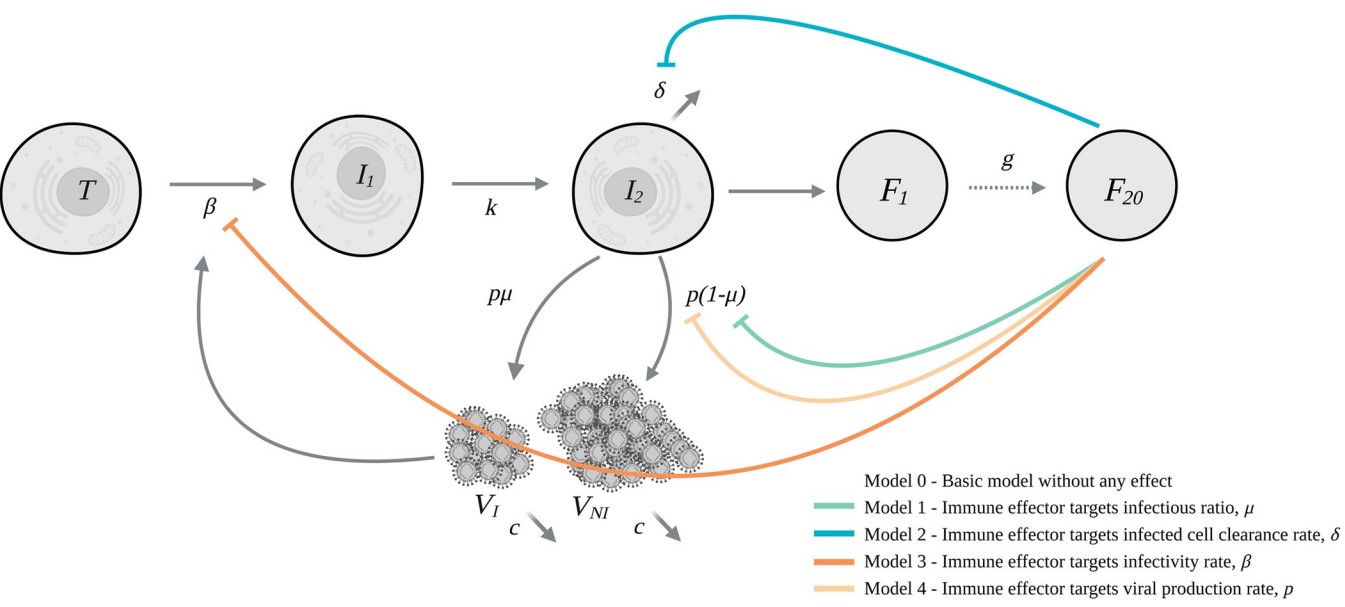

**Fig 2. Schematic model of SARS-CoV-2 infection and action of the immune system.** The basic model is a target cell limited model without any immune response. The parameters are: $\beta$ the infectivity rate, $k$ the transfer rate between non-productive and productive infected cells, $\delta$ the loss rate of productive infected cells, $p$ the viral production rate, $\mu$ the ratio of infectious virus, $g$ the transfer rate between the compartments of the immune response and $c$ the loss rate of both infectious and non-infectious virus.

total number of infectious virus produced by one cell over its lifespan at the beginning of the infection) of 136 (95% CI 121−153).

## VoC specific effect on viral dynamic parameters

Once an effect of the immune response was selected, a covariate search algorithm was used to find the most likely VoC associated effects (see methods) and considered the historical variant as the reference. Several variant-specific covariates were found on viral kinetics parameters that we detail below (Fig 3 and S2 Table). First, beta variant was characterized with a reduced infected cells death rate ($\delta$) by a factor of 0.7 (95% CI 0.6−0.9) compared with the historical variant (p-value < 0.01). This led to an infected cell half-life of 17 hours and resulted in a longer period of viral load shedding as infected cells produced viruses for longer period of time. Gamma variant had an effect on the parameter $\theta$ (p-value < 0.001), the amount of immune effector $F_{20}$ required to reduce by half the infectious ratio, increasing it by a factor of 9508 (95% CI 387−50 041) resulting in higher peak viral load and a longer duration of infectious virus shedding (Fig 4). Variant delta is characterized by an effect on both $\theta$ (p-value < 0.001) and the viral production parameter $p$ (p-value < 0.05), increasing those parameters by factors

**Table 1. Alternative immune response models.**

| Models | Description | ΔBIC |
|---|---|---|
| Reference model | Absence of immune response | − |
| **Model 1** | **Reduction of the infectious ratio** | **−43** |
| Model 2 | Increase in infected cell clearance | −15 |
| Model 3 | Reduction of viral infection rate | −36 |
| Model 4 | Reduction of the viral production | +9 |
| Model 5 | Cells become refractory to infection | -15 |

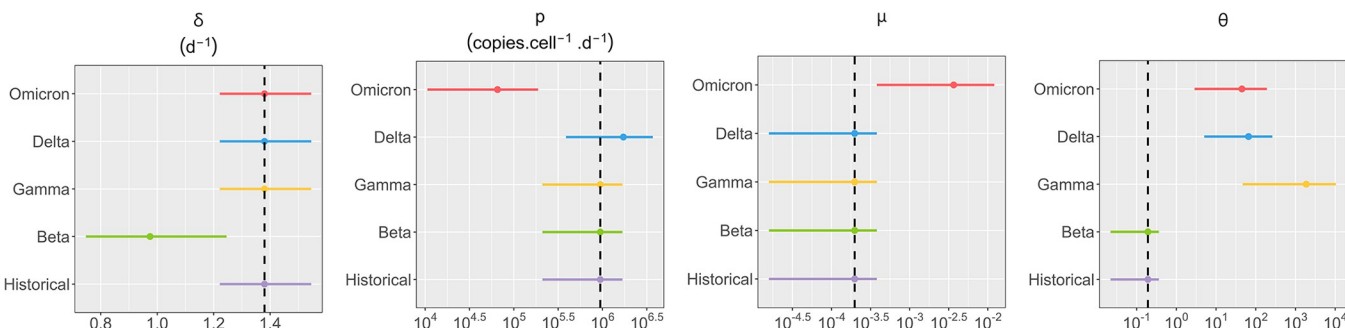

**Fig 3. Estimated population parameters for each variant.** We represent the mean value and 95% confidence interval of populations parameters for each variant. We represent only parameters having at least one variant-specific effect. Full table for population parameters is in S2 Table. The dashed black line represents the historical value.

336 (95% CI 49–1191) and 1.78 (95% CI 1–3) respectively. Finally, omicron variant (BA.1) affected the parameters of the immune system $\theta$ (p-value < 0.001), the viral production rate parameter $p$ (p-value < 0.001) and the infectious ratio $\mu$ (p-value < 0.001) modifying them by factors 229 (95% CI 27–884), 0.07 (95% CI 0.02–0.2) and 18 (95% CI 4–51) respectively (Fig 4). The model well reproduced the viral load of all animals in the individuals fits (S2 Fig). To challenge our model assumption on the immune response compartment, we performed a sensitivity analysis on the number of transitioning compartments ($j$) and the mean time spent in those compartments ($\tau$). We also performed the covariate selection on all models. We found that largely similar VoC-specific covariates were selected regardless of the delay in the immune response (S3 and S4 Figs). In addition, we found that the model assuming a time spent of 3 days in the transitioning compartment yielded the best results (S3 Table).

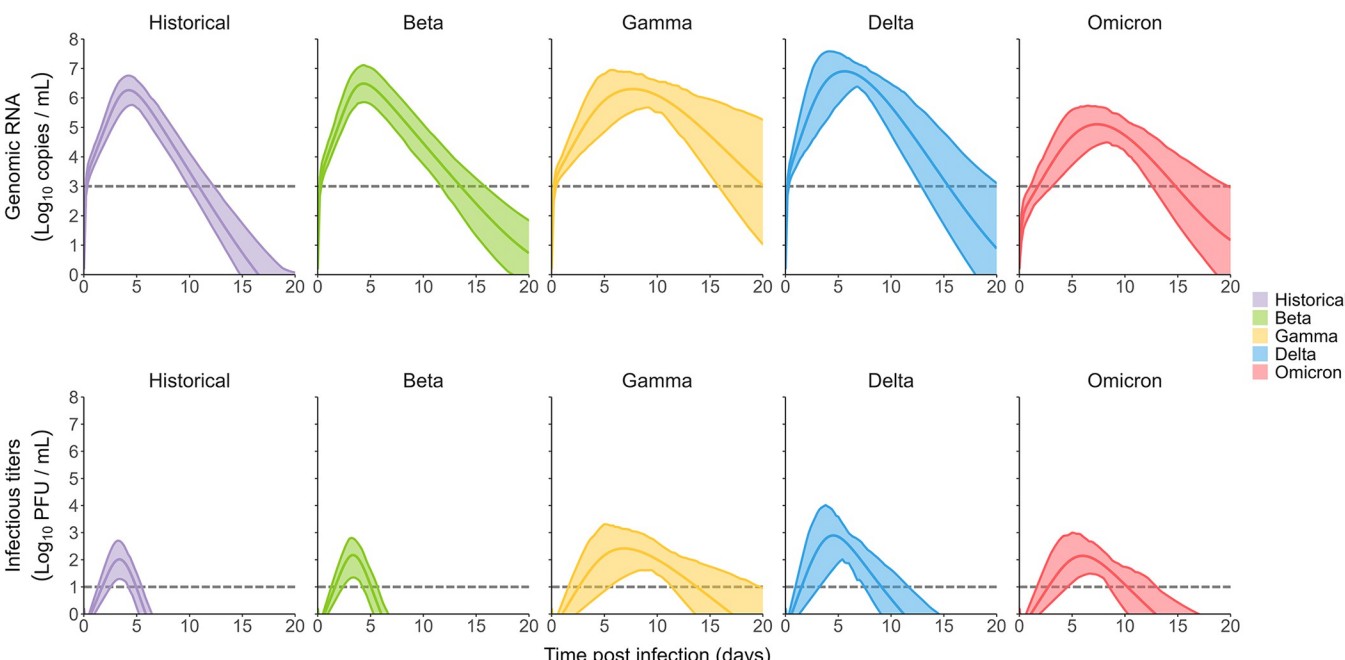

**Fig 4. Simulation of variant of concern impact on viral load.** Using simulations, we sampled parameters considering both the uncertainty in the estimation and the inter-individual variability (see methods). We represent the mean viral load of all variants and its 95% confidence interval. Dotted lines are the limits of detections.

### Predicted impact of variants in a natural infection setting

The main limitation of translating these results to humans is the fact that infection in animals is done with a large inoculum dose ($10^5$–$10^6$ PFU), while human infections are presumably initiated with much lower virus dose [18]. Human experimental infections were performed with 10 $TCID_{50}$ [19] in the nose, i.e., 10,000–100,000 times less virus than in the animal model. Using simulations with a lower inoculum of 1, 10 and 100 PFU (see methods), we are able to derive metrics of interest for each variant. The results obtained between 1 PFU are identical to those observed with 10 PFU, while 100 PFU predicts a more rapid kinetics (S5 Fig). We present the results with 10 PFU in the following.

The historical variant is characterized by a mean time to peak of 4.3 dpi (95% CI 3.7–4.8) and of 3.5 dpi (95% CI 3–3.9) for genomic RNA and infectious titers respectively. We found a mean peak viral load of 6.3 $\log_{10}$ copies/mL (95% CI 5.5–7) and of 2.1 PFU/mL (95% CI 1.2 –2.9) for genomic RNA and infectious titers, respectively.

The reduced infected cell clearance rate of the beta variant resulted in a longer period of viral load shedding. The duration of the acute infection stage was consequently increased from 10.9 days (95% CI 9.5–13.1) for the historical variant to 13.4 days (95% CI 11.1–15.7) for the beta variant.

All variants except beta have shown an effect on the antigen-mediated response, greatly reducing its impact on viral kinetics. As the effect of the antigen-mediated response was reduced, the infectious ratio was increased leading to more infectious particles produced over longer periods of time. This led to the increase of the infectious titers clearance stage duration from 1.5 days for the historical variant (95% CI 0.6–1.9) to 6 days (95% CI 4.4–7.5), 3.8 days (95% CI 3.1–4.6) and 3.7 days (95% CI 2.8–4.5) for the gamma, delta and omicron variants respectively (Fig 5). This is in line with numbers of studies showing the immune escape capabilities of those variants [20–22].

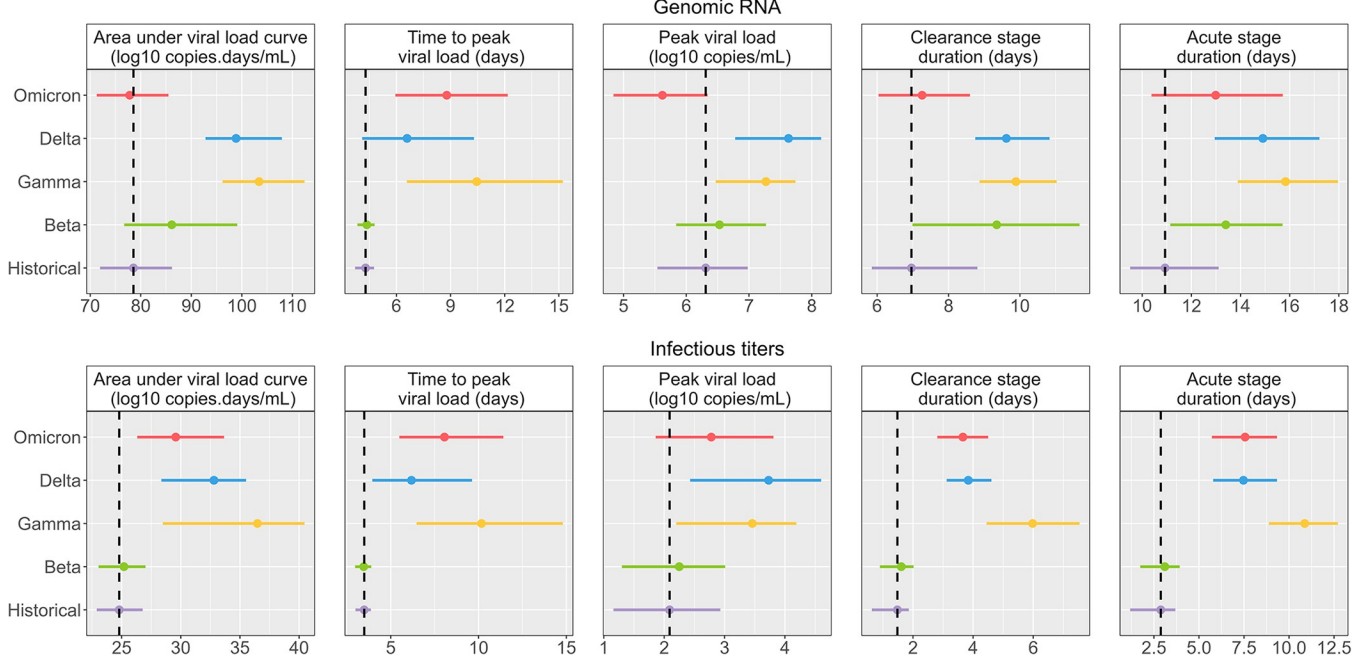

**Fig 5. Impact of VoC on viral load metrics in the context of an infection with a low inoculum.** We represent the mean and 95% confidence interval for each variant. The dashed black line represents the historical mean value.

An effect increasing the viral production parameter ($p$), as observed for the delta variant, results in largely higher peak viral load of 7.6 $\log_{10}$ copies/mL (95% CI 6.8–8.2) and peak infectious titers of 3.7 PFU/mL (95% CI 2.4–4.6). Conversely, an effect reducing the viral production parameter, as observed for the omicron variant, results in lower peak viral load compared to the historical variant of 5.6 $\log_{10}$ copies/mL (4.8–6.3) but very similar peak infectious titers at 2.8 PFU/mL (95% CI 1.9–3.8). This is due to an effect of omicron on the infectious ratio, increasing the proportion of infectious virus produced.

## Discussion

Here, we used mathematical models to characterize in detail the SARS-CoV-2 viral dynamics with the main variants of concern, using data obtained in an experimental model of non-human primates. We evaluated the impact of an antigen-mediated immune response on the viral dynamics and found that an effect reducing the infectious ratio best described our data. Gamma, delta and omicron (BA.1) variants showed a significant ability to escape this response, leading to a higher ratio of infectious virus than the historical variant, especially at later time-points. Interestingly, delta variant was associated with an increased viral production rate and therefore higher viral loads, whereas omicron variant was associated with a lower viral production rate. Altogether, our model predicts that omicron infections are associated with lower peak viral RNA and reduced duration of viral RNA clearance compared to delta variant, while their kinetics of infectious titers are similar. Accordingly, it suggests that the increased transmissibility of omicron variant is not caused by increased viral load, but rather to its ability to maintain high levels of infectious particles, and may suggest longer duration of infectiousness for both delta and omicron variants. They illustrate that the quantification of infectious titers over time is crucial to inform further public health policies and adjust the isolation period accordingly [23].

Our study has important limitations that need to be acknowledged. First, in this experimental model the inoculated dose is extremely high compared to a natural infection, which is typically initiated with 1–10 infectious particles (19). This is inherent limitation of NHP studies leads to a rapid saturation of target cells making it difficult to accurately estimate the early phase of infection. Here, however, our prediction obtained assuming a lower inoculum is in line with what was observed in humans, with a peak viral load around 5 days post infection [24,25]. In the future, studies evaluating viral dynamics with lower inoculum could be helpful to tease out a potential impact of the inoculum size on viral dynamics. Second, as our model selection process was performed using all data available, it assumes that the same model applies to all variants, which may not be true. Nonetheless, data limitation did not allow to perform model selection on each variant separately. Third, we developed an extension of the target cell limited model considering the effect of an antigen-mediated immune response targeting the infectious ratio parameter μ. This effect on the infectious ratio could be attributable to an effect of the innate response, in particular the interferon (IFN) response, that can confer antiviral, antiproliferative and immunomodulatory functions [26]. Our model also identified that the effects of the immune response was variant-dependent (captured by the covariates on the parameter $θ$ in the model) consistent with the observations that SARS-CoV-2 variants exhibits mutations in nucleocapsid, membrane and non-structural proteins that antagonize IFN signalling in cells [27,28]. Unfortunately, this hypothesis remains speculative as IFN was not measured in this study, and the cytokines that were measured (e.g. MCP1, IL15 and IL1RA) did not correlate with the viral kinetics (S7 Fig). Of note, these cytokines peaked at day 2, i.e., earlier than predicted by our model which assumes a peak of the immune response by day 6 (S6 Fig). This could be due to the fact that these cytokines were measured in plasma, and not

in the site of infection, but we nevertheless tested the sensitivity of this prediction to our hypotheses. For that purpose, we conducted a sensitivity analysis on the kinetics of the immune response by assuming different number of compartments ($j$) and mean time spent in those compartments ($\tau$), allowing to modulate the mean delay of immune response and its distribution. Interestingly, in all simulations, the model predicted a peak of the immune response later than day 4. Using the final model, we nevertheless show that the impact of the immune response in reducing the infectious ratio is already significant at day 3 post infection (S6B Fig).

Regarding virus infectivity, several aspects need to be discussed regarding both their measurements and their interpretation. First, infectious titers are only a measure of *in vitro* infectivity, and to what extent they translate into infectiousness is unknown. Second, recent results obtained with other experimental cell line suggest that Vero E6 cells may underestimate viral production of Omicron-variants as represented by the parameter $\theta$ [29]. Related to that question, one should acknowledge that the amount of infectious virus remains limited, both in terms of number of points and in the capability to measure it precisely. Although the data available here allowed us to estimate all parameters with good precision, including the changes in infectivity over time, some aspects of the model (such as the effect of the immune system on infectivity, noted $\theta$) remain phenomenological. Finally, in a context where more than half of the world population has received at least one dose of COVID-19 vaccine [30], and probably even more have been infected by one or different variants, our results will need to be complemented by studies to evaluate the impact of vaccination, previous infection or both on viral dynamics.

## Materials and methods

### Experimental procedure

Data comes from studies performed on cynomolgus macaques to evaluate the viral dynamics of SARS-CoV-2 variants. Our study includes 78 cynomolgus macaques (*Macaca fascicularis*) coming from control arms of several studies and have received no pharmacological interventions besides placebo. All animals were infected with doses ranging from $7 \times 10^4$ to $10^6$ PFU of different SARS-CoV-2 strains. Animals are infected via both nasopharyngeal and intratracheal route with 10% of the initial volume administered in the nose and 90% in the trachea. The study is composed of 5 groups, each infected with a different SARS-CoV-2 strains: 44 Historical (hCoV-19/France/lDF0372/2020 strain; GISAID EpiCoV platform under accession number EPI_ISL_406596), 9 Bêta (B.1.351—hCoV-19/USA/MD-HP01542/2021, BEI NR-55283), 5 Gamma (P.1 - hCoV-19/Japan/TY7-503/2021, BEI NR-54984), 11 Delta (B.1.617.2—hCoV-19/USA/MD-HP05647/2021, BEI NR-55674) and 9 Omicron (B.1.1.529 –hCoV-19/USA/MD-HP20874/2021, BEI NR-56462). For each group both genomic RNA and subgenomic RNA swab samples were quantified using real time PCR in both the nasopharynx and in the trachea. For 41 animals (13 Historical, 3 Beta, 5 Gamma, 11 Delta and 7 Omicron (BA.1)) infectious titers were measured at 2 time points, early (2, 3 or 4 days post infection) and late (5 or 7 days post infection) using Tissue Culture Infectious Dose ($TCID_{50}$) from nasopharyngeal swab samples [16]. As we included animals from different studies that were inoculated with different methods (PFU or $TCID_{50}$), we normalized all measures of infectious titers by converting all $TCID_{50}$ measurements to Plaque Forming Units (PFU) using the Formula 1 PFU = 0.7 $TCID_{50}$ [17]. As no infectious titers were measured in the trachea samples, we focused the main analysis on the nasopharyngeal compartment. The results mainly focus on the genomic viral load as the subgenomic is a directly proportional to the latter.

## Basic viral dynamic model

We used a previously described model of SARS-COV-2 viral dynamics to reconstruct the nasopharyngeal viral load of infected animals. In this model, target cells (T) become infected cells ($I_1$) at a rate $\beta$. Infected cells transition into productive infected cells ($I_2$) at a rate $k$ and produce infectious virus ($V_I$) at a rate $p\mu$ and non-infectious virus ($V_{NI}$) at a rate $p(1-\mu)$. Productive infected cells are cleared at a rate $\delta$ and both infectious and non-infectious virus are cleared at a rate $c$. The basic within-host reproductive number, representing the number of newly infected cells by one infected cell, is $R_0 = \frac{\beta p T_0 \mu}{c\delta}$ and the burst-size, representing the number of infectious virus produced by one infected cell over its lifespan, is $N = \frac{p\mu}{\delta}$. The model is described with the following set of ordinary differential equations:

$$\frac{dT}{dt} = -\beta V_I T \tag{1}$$

$$\frac{dI_1}{dt} = \beta V_I T - kI_1 \tag{2}$$

$$\frac{dI_2}{dt} = kI_1 - \delta I_2 \tag{3}$$

$$\frac{dV_I}{dt} = p\mu I_2 - cV_I \tag{4}$$

$$\frac{dV_{NI}}{dt} = p(1-\mu)I_2 - cV_{NI} \tag{5}$$

## Assumption on parameter values

Some parameters of the model were fixed to ensure identifiability. The transfer rate parameter between infected cells and productive infected cells was fixed to $k = 4$ day$^{-1}$ (corresponding to a mean duration of the eclipse phase, i.e. the time for infected cells to start producing viruses, of $\frac{1}{k} = 6$ hours) [31]. The viral clearance c was set to 10 day$^{-1}$ based on previous work [14,16,24]. As only the product $pT_0$ is identifiable, we choose to fix the initial number of target cell to $T_0 = 12\ 500$ cells following the same assumptions as in [16]. Animals are infected using both an intranasal and intratracheal route using 90% of the initial inoculum in the trachea and 10% in the nose [13]. We take this into account by adding a 0.1 factor in our initial conditions. We introduced a parameter $h$ representing the proportion of the inoculum that arrive on the site of infection. Because this parameter cannot be reliably estimated or experimentally measured, a profile likelihood approach was used, testing values ranging from 10% to 90%. Therefore, our initial conditions were set to:

$$T_0(t=0) = 1.25 \times 10^4 \tag{6}$$

$$I_1(t=0) = 0 \tag{7}$$

$$I_2(t=0) = 0 \tag{8}$$

$$V_I(t=0) = V_I(0)_i \times h_i \times 0.1 \tag{9}$$

$$V_{NI}(t=0) = V_{NI}(0)_i \times h_i \times 0.1 \tag{10}$$

Where $V_I(0)_i$ is the number of infectious virus of subject $i$ (obtained assuming that 1 PFU = 1 infectious particle), $V_{NI}(0)_i$ is the number of non-infectious virus (obtained by the difference between the total number of RNA copies and the number of infectious titers in the inoculum) and $h_i$ the proportion of the inoculum actively initiating the infection.

## Models incorporating antigen-mediated immune response

To account for the quick drop in infectious titers observed for the historical variant (Figs 1 and S1), we introduced a model incorporating an action of an antigen-mediated immune response. To allow some variability in this response we assumed a certain production phase before the immune response takes effect. We introduced this delayed effector compartment using the Linear Chain Trick (LCT) [32] by modelling a successive number of transitioning compartments. This assumes an Erlang distribution with parameters $j$ and $\tau$ representing the number of transitioning compartments and the mean time spent in those compartments respectively. We fixed those parameters to $j = 20$ compartments and $\tau = 3$ days to account for the setup of the immune response. We then performed a sensitivity analysis on those parameters varying both the number of compartments (from 5 to 30) and the mean time spent in those compartments (from 1 day to 6).

The equations for the transfer compartments are written as follows:

$$\frac{dF_1}{dt} = I_2 - gF_1 \tag{11}$$

$$\frac{dF_2}{dt} = gF_1 - gF_2 \tag{12}$$

$$\frac{dF_{20}}{dt} = gF_{19} - d_F F_{20} \tag{13}$$

In the following only the compartment $F_{20}$ will serve as the effector for the action of the immune system. The transfer rate parameter $g$ is then written as $\frac{j}{\tau}$ and fixed to 6.67 d$^{-1}$ and the loss rate of the final effector $d_F$ is fixed to 0.4 d$^{-1}$ [24]. All immune response compartments were set to 0 at t = 0. Several modes of action of the response system were tested:

Model 1: Immune effector decreases the infectious ratio $\mu$

In this model, the immune effector directly decreases the infectious ratio parameter $\mu$ using an Emax function type expression:

$$\frac{dV_I}{dt} = p\mu\left(1 - \frac{F_{20}}{F_{20} + \theta}\right)I_2 - cV_I \tag{14}$$

$$\frac{dV_{NI}}{dt} = p\left(1 - \mu\left(1 - \frac{F_{20}}{F_{20} + \theta}\right)\right)I_2 - cV_{NI} \tag{15}$$

With $\theta$ being the amount of immune effector $F_{20}$ needed to reduce by half the infectious ratio.

Model 2: Immune effector increases infected productive cells death rate $\delta$

The death rate of infected cells is increased in proportion to the amount of immune effector $F_{20}$.

$$\frac{dI_2}{dt} = kI_1 - \delta(1 + \varphi F_{20})I_2 \tag{16}$$

Where $\varphi$ is the strength of the immune system.

<u>Model 3: Immune effector reduces the infectivity rate $\beta$</u>

In this model, the immune effector blocks virus entry in the cells by reducing the infectivity parameter $\beta$.

$$\frac{dT}{dt} = -\beta\left(1 - \frac{F_{20}}{F_{20} + \theta}\right)V_I T \tag{17}$$

$$\frac{dI_1}{dt} = \beta\left(1 - \frac{F_{20}}{F_{20} + \theta}\right)V_I T - kI_1 \tag{18}$$

<u>Model 4: Immune effector reduces the production rate $p$</u>

In the same way as model 1, the viral load production parameter is reduced by the immune effector with an Emax type function:

$$\frac{dV_I}{dt} = p\left(1 - \frac{F_{20}}{F_{20} + \theta}\right)\mu I_2 - cV_I \tag{19}$$

$$\frac{dV_{NI}}{dt} = p\left(1 - \frac{F_{20}}{F_{20} + \theta}\right)(1 - \mu)I_2 - cV_{NI} \tag{20}$$

All models were compared based on the Bayesian Information Criterion (BIC). We selected the model that yielded the lowest BIC and the best individual fits.

<u>Model 5: Immune effector induce refraction to infection.</u>

The immune response renders target cells refractory to infection [24] as follows:

$$\frac{dT}{dt} = -\beta T V_I - \varphi \frac{F_{20}}{F_{20} + \theta} T \tag{21}$$

Models were estimated using all data (historical and variants) and compared based on BIC, precision of the estimation and goodness of fit. A covariate search algorithm was then used on the selected model (see below) to identify VoC-specific effects.

## Statistical model

Parameter estimation was performed using non-linear mixed effect modelling. The statistical models describing the genomic RNA, subgenomic RNA and the infectious titers are:

$$y_{ij}^1 = \log_{10} V(t_{ij}, \Psi_i) + \epsilon_{ij}^1 \tag{22}$$

$$y_{ij}^2 = \log_{10} f \times I_2(t_{ij}, \Psi_i) + \epsilon_{ij}^2 \tag{23}$$

$$y_{ij}^3 = \log_{10} V_I(t_{ij}, \Psi_i) + \epsilon_{ij}^3 \tag{24}$$

Where the superscript 1, 2 and 3 refers to the genomic RNA, subgenomic RNA and infectious titers, respectively. We denote $y_{ij}$ is the $j^{th}$ observation of subject $i$ at time $t_{ij}$, with $i \in 1,$ ..., N and $j \in 1, \ldots, n_i$ with N the number of subject and $n_i$ the number of observations for

subject $i$. The function describing the total viral load kinetics $V(t_{ij}, \Psi_i)$ predicted by the model at time $t_{ij}$ defined as: $V_I(t_{ij}, \Psi_i)+V_{NI}(t_{ij}, \Psi_i)$ predicted by the model at time $t_{ij}$. The vector of individual parameters of subject $i$ is noted $\Psi_i$ and $\epsilon_{ij}$ is the additive residual Gaussian error of constant standard deviation $\sigma$. The vector of individual parameters depends on a fixed effects vector and on an individual random effects vector, which follows a normal centered distribution with diagonal variance-covariance matrix $\Omega$. All parameters follow a log-normal distribution to ensure positivity except both parameters $\mu$ and $h$ which follows logit-normal distribution and are bounded between 0 and 1. We assumed random effect on all parameters and removed them using backward procedure, if they were < 0.1 or their RSE > 50%. All biomarkers (i.e. genomic RNA, subgenomic RNA and infectious titers) were fitted simultaneously.

## Selection of variant-specific effect on the viral dynamic parameters

Using the best model selected at the previous step, we sought to identify VoC-specific effect on the parameters of the model ($\beta$, $\delta$, $p$, $\mu$ and $\theta$). We first performed a backward selection of the random effects removing non-significant ones (i.e. relative standard error > 50%) if the BIC wasn't degraded by more than 2 points. We then used the Conditional Sampling use for Stepwise Approach on Correlation tests (COSSAC) to identify variant specific effect [33]. Then a backward procedure was used to remove any non-significant covariate effect with a Wald test (i.e. the covariate was removed if its coefficient effect relative standard error was > 50%). This procedure was repeated until all nonsignificant covariate effects had been eliminated. Additionally, we performed a sensitivity analysis on our best structural model. We tested for several delays in the establishment of the antigen-mediated effector (from 1 to 6 days) and on the number of transitions compartments (from 5 to 30) and then performed the covariate search on all model combinations.

## Simulation of natural human infection

Finally, we used our final model to assess the impact of variants of concern on viral load and viral infectivity in a natural infection setting. We used a starting inoculum of 10 infectious virus, as described in an experimental challenge conducted in England [19] to simulate a human infection. The initial conditions are then written as:

$$V_I(t = 0) = 10$$

$$V_{NI}(t = 0) = 0$$

We provided confidence interval on the mean predicted viral load, considering both the uncertainty in the estimation and the inter-individual variability. We first sampled M = 100 population parameters in their estimation distribution and then, for each variant, sampled N = 30 individual parameters from each sets of population parameters (leading to 3000 individual parameters per variant). We calculated the predicted viral load of all individuals and derived the mean viral load over the simulated individuals at all times with its 95% inter quantile range. Additionally, we provided the distribution of several viral dynamic metrics, namely:

- the area under viral load curve,

- the peak and time to peak viral load

- the duration of the clearance stage, calculated as the time interval between the peak viral load and the time to undetectable viral load

- the duration of the acute phase, calculated as the time between the first and the last detectable viral load [34].

Additional simulations were done with different inoculum (1 and 100) to assess sensibility to initial conditions.

## Parameter estimation

All parameters were estimated by computing the maximum-likelihood estimator using the stochastic approximation expectation-maximization (SAEM) algorithm implemented in Monolix Software 2020R1 [35,36]. Standard errors and the likelihood were computed by importance sampling.

## Supporting information

**S1 Fig. Relationship between genomic RNA and infectious titers.** We represent the longitudinal values of genomic RNA for each individual and if the associated PFU sampe is detectable or not.
(PDF)

**S2 Fig. Individual fit of genomic RNA, subgenomic RNA and infectious titers in all animals.** Undetectable values are represented as empty dots. Values above the upper limit of quantification are represented as squares.
(PDF)

**S3 Fig. Sensitivity analysis on the covariate selection algorithm.** We performed a sensitivity analysis on our best model. The model IDs are represented on top, as described in S3 Table. The scale represents the magnitude of the covariate effect rescaled for each row with 0 being the minimum value and 1 the maximum. Empty tiles indicate that no covariates were selected for this variant-parameter relationship.
(PDF)

**S4 Fig. Consistency of the covariate selection algorithm.** We represent the number of times a covariate was found on a variant-parameter relationship across all 24 models. Empty tiles indicate that no covariates were found for this variant-parameter relationship.
(PDF)

**S5 Fig. Impact of different inoculum on viral dynamic simulations.** Using simulations, we sampled parameters considering both the uncertainty in the estimation and the inter-individual variability (see methods). Only the mean viral load was shown for clarity.
(PDF)

**S6 Fig. Dynamics of the immune response and its effect.** A) Median trajectory of the last compartment of our immune response B) Median trajectory of the infectious ratio parameter $\mu$ over time. We used the population parameters of our best model to simulate the median trajectory of each variant.
(PDF)

**S7 Fig. Dynamics of cytokines and correlation with viral load metrics.** A) Median concentration of measured cytokines. B) Correlation between AUC of viral load predicted by our model and the cytokine AUC. C) Correlation between peak viral load predicted by our model and peak cytokine concentration.
(PDF)

**S1 Table. Characteristics of the 78 animals analysed.** Descriptive statistics of the animals calculated on the raw data.
(DOCX)

**S2 Table. Estimates of the population parameter and covariate effects for the best model.** **The standard error for the $R_0$ parameters were calculated using the delta method.
(DOCX)

**S3 Table. Sensitivity analysis on the delayed immune response.** Using the best structural model (i.e. Model 1 including an effect on the infectious ratio) we tested several delays for the immune response to take place and performed the covariate search algorithm on all models.
(DOCX)

## Acknowledgments

We would like to thank everyone in the CEA and at Pasteur Institute that have helped for data collection. We thank Alan Perelson for helpful discussions.

## Author Contributions

**Conceptualization:** Aurélien Marc, Romain Marlin, Flora Donati, Mélanie Prague, Marion Kerioui, Cécile Hérate, Marie Alexandre, Julie Bertrand, Vanessa Contreras, Sylvie Behillil, Pauline Maisonnasse, Sylvie Van Der Werf, Roger Le Grand, Jérémie Guedj.

**Data curation:** Romain Marlin, Flora Donati, Cécile Hérate, Nathalie Dereuddre-bosquet, Sylvie Behillil, Pauline Maisonnasse, Sylvie Van Der Werf, Roger Le Grand.

**Formal analysis:** Aurélien Marc, Mélanie Prague, Marion Kerioui, Marie Alexandre.

**Funding acquisition:** Jérémie Guedj.

**Investigation:** Romain Marlin, Flora Donati, Cécile Hérate, Vanessa Contreras, Sylvie Behillil, Pauline Maisonnasse, Sylvie Van Der Werf, Roger Le Grand, Jérémie Guedj.

**Methodology:** Aurélien Marc, Romain Marlin, Mélanie Prague, Marion Kerioui, Cécile Hérate, Marie Alexandre, Julie Bertrand, Vanessa Contreras, Jérémie Guedj.

**Project administration:** Sylvie Van Der Werf, Roger Le Grand, Jérémie Guedj.

**Software:** Aurélien Marc, Mélanie Prague, Marion Kerioui, Marie Alexandre, Julie Bertrand.

**Supervision:** Nathalie Dereuddre-bosquet, Sylvie Behillil, Sylvie Van Der Werf, Roger Le Grand, Jérémie Guedj.

**Visualization:** Aurélien Marc.

**Writing – original draft:** Aurélien Marc, Marion Kerioui, Jérémie Guedj.

**Writing – review & editing:** Aurélien Marc, Romain Marlin, Flora Donati, Mélanie Prague, Marion Kerioui, Cécile Hérate, Marie Alexandre, Julie Bertrand, Jérémie Guedj.

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
