## [Decision Letter · Decision Letter 0]

28 Dec 2022

Dear Student MARC,

Thank you very much for submitting your manuscript "Impact of variants of concern on SARS-CoV-2 viral dynamics in non-human primates." for consideration at PLOS Computational Biology.

As with all papers reviewed by the journal, your manuscript was reviewed by members of the editorial board and by several independent reviewers. In light of the reviews (below this email), we would like to invite the resubmission of a significantly-revised version that takes into account the reviewers' comments.

We cannot make any decision about publication until we have seen the revised manuscript and your response to the reviewers' comments. Your revised manuscript is also likely to be sent to reviewers for further evaluation. However I do feel the comments can be addressed.

Sincerely,

Rustom Antia

Academic Editor

PLOS Computational Biology

Amber Smith

Section Editor

PLOS Computational Biology

Reviewer's Responses to Questions

**Comments to the Authors:**

Reviewer #1: The authors used longitudinal viral load data from macaques infected with SARS-CoV-2 to study the viral dynamics of some SARS-CoV-2 variants. They analyzed the data using several hypothesized mathematical models to understand how the virus infection dynamics differ among the major variants of concern (VOCs). They found that Omicron variants had a high infectious virus production rate but a low peak viral load, and they discussed the relationship between these features and the high infectivity and low pathogenicity of Omicron variants. However, the current version of the manuscript may lack a discussion of immunity, despite the fact that the conclusion of the research is closely related to the host immune response. Also the mathematical modeling and data analysis are need to be update. Therefore, I recommend resubmitting the revision with the following concerns addressed:

1. First of all, the data for TCID50 is extremely limited, with only two time points at maximum and many undetectable values, particularly in the late phase. Despite using these limited datasets, the authors concluded that the best model among four hypothetical mechanisms of action by immune cells is a model targeting the infectious ratio through immune response (Model 1). Apart from BIC, they need to explain how this model is validated from immunological point of view rather than statistical point of view.

2. In Guedj et al. (PMID: 33536313), they proposed “refractory model” and use it for analyzing the longitudinal VL data. What is the reason that they do not propose the refractory model here for the candidate model. In fact, there are a hundred of different mathematical model considering different types of immune responses?

3. They chose the Model 1 by data fitting to “historical variant” and then a covariate search algorithm was used to find the most likely VOC associated effects. In general, because SARS-CoV-2 evolves to escape host immune response including vaccine-elicited immune responses, the mechanism of action by immune system may be different. That is, even if the best model for historical variants is Model 1, best models for other VOCs might be different model (e.g., Model 2, Model 3, Model 4). In addition, they need to explain how VOC-specific mutations in sequences is corresponding to an ability to escape from immune responses (i.e., the mutations on VOCs affect the function of the infectious virus production).

4. While estimated parameter on mu (i.e., the ratio of infectious virus), there is no discussion on theta (i.e., the amount of immune effector needed to reduce by 50% infectious ratio). In terms of data fitting, independently estimation mu and theta are difficult because these two parameters have a complementary role for virus infection dynamics (i.e., distinguish [small mu and large theta] and [large mu and small theta] is difficult).

5. They found “All variants except beta have shown an effect on the antigen-mediated response, greatly reducing its impact on viral kinetics. As the effect of the antigen-mediated response was reduced, the infectious ratio was increased leading to more infectious particles produced over longer periods of time.” and explained “This is in line with numbers of studies showing the immune escape capabilities of those variants”. The authors have to discuss how their conclusion based on estimated parameters for immune escape (i.e., changes on mu and theta) are consistent with the current knowledge on SARS-CoV-2 immune escape because, to date, there are numerous advances on understanding immune response to different VOCs.

6. The authors have not included a simulation of the host immune response by their mathematical model. In order to validate their mathematical model and the estimated parameters, they should present a simulation of the host immune response to VOCs and discuss how it compares to recent studies on immune responses to VOCs.

7. The assumptions about the initial values are unclear. What do "0.1" and "V_NI(0)_i-V_I(0)_i" represent? It is not clear if it is appropriate to perform arithmetic operations with these values, given that they have different units.

8. It is not clear what the purpose of fitting the subgenomic RNA is or if it improves the parameter estimation.

9. As they also mentioned in Discussion section, the initial virus dose is much higher than natural infection, so the estimation of the parameters related to the growth phase of virus production may not be accurate. Especially, this is expected to have affected the time-to-peak and duration of virus shedding of estimated viral dynamics. The effect of the initial virus dose on parameter estimation should be discussed.

10. The authors assumed that the correction for the amount of measured viral load due to the structure of the animal's nasal cavity is 20% without providing any justification or references. This assumption should be supported with references.

11. In Fig S2, cases 77 and 78 had poor fitting results because there were not enough detectable points. This may significantly impact the parameter estimation for Omicron variants and the overall features of viral dynamics. Is there a specific reason why they were not excluded?

Reviewer #2: The study investigates virus and immune dynamics in 78 macaques challenged with the original strain and four variants of concern with the aim of determining each strain viral-immune-specific characteristics. The study is clearly reported, the results are interesting, and the scope is appropriate for the journal. I have a few concerns/suggestions that I will detail below

1. Please add a parameter table and include initial condition. Some IC are listed throughout the text but I could not find initial conditions for F1, …, F20.

2. I am surprised about the choice of VI(0). While I understand that it was set at 10 to match human studies, it is very low compared to the inoculum RNA. Do the results change if VI(0) is varied? Given that the study investigates the differences in VoC, it is possible that the initial inoculum is a factor as well. Maybe fewer VI start infection in certain variants. Can you add VI(0) as an unknown in the data fitting?

3. Cn you explain the reasoning for equating I2 with the subgenomic data. Why not I1+I2?

4. More details are needed for the methodology:

a. how are the three data sets weighted?

b. Which procedure was used for sensitivity?

c. What determined the choice of distributions for different parameters, and are those choices biasing the results?

5. All longitudinal data should be presented in the manuscript. Also, it is very hard to distinguish between squares and circles, maybe color (or filled objects) will make that easier to understand.

6. Why are some F effects density dependent? Have the authors investigate linear and power law terms?

**Have the authors made all data and (if applicable) computational code underlying the findings in their manuscript fully available?**

Reviewer #1: **No: **No data for FigS2 and code for the fitting.

Reviewer #2: **No: **I did not see any place where data or code was shared. Moreover, the longitudinal data should be shared so the study can be reproduced.

PLOS authors have the option to publish the peer review history of their article (what does this mean?). If published, this will include your full peer review and any attached files.

Reviewer #1: No

Reviewer #2: No
---

## [Decision Letter · Decision Letter 1]

21 May 2023

Dear Student MARC,

Thank you very much for submitting your manuscript "Impact of variants of concern on SARS-CoV-2 viral dynamics in non-human primates." for consideration at PLOS Computational Biology. As with all papers reviewed by the journal, your manuscript was reviewed by members of the editorial board and by several independent reviewers. The reviewers appreciated the attention to an important topic. Based on the reviews, we are likely to accept this manuscript for publication, providing that you modify the manuscript according to the review recommendations.

This is a minor but required revision. Please explicitly address the concern of Reviewer 1. With that addressed I believe this will be a very nice paper in PLOS comp bio.

Sincerely,

Rustom Antia

Academic Editor

PLOS Computational Biology

Amber Smith

Section Editor

PLOS Computational Biology

This is a minor but required revision. Please explicitly address the concern of Reviewer 1. With that addressed I believe this will be a very nice paper in PLOS comp bio.

Reviewer's Responses to Questions

**Comments to the Authors:**

Reviewer #1: Regarding my previous comment (Comment 1), the authors hypothesized in their revision that the immune response modeled in this mathematical model would predominantly favor the effect of the innate immune response, which serves as the host's first line of defense during an infection. However, the authors were unable to identify a clear correlation between the model-predicted immune response (F20) in S6 Fig and the measured cytokines (IL15, IL1ra, MCP1) in S7 Fig, as they did not measure IFN responses. Notably, the measured cytokine peaks occur around 2 dpi, while the model prediction peak occurs after 5 dpi. This discrepancy arises because the authors assumed a delay of 3 days in the antigen-mediated immune induction (Eqs. 11-30), which implies that the peak cannot occur before 3 dpi. If the authors wish to reference the measured cytokine dynamics, they must relax the assumption of fixed parameters in the antigen-mediated immune induction process of Model (1), specifically j, tau, and d_F. Otherwise, I fail to comprehend the purpose of including and comparing S6 Fig and S7A Fig as conducted by the authors, and I anticipate that validating their hypothesis would be challenging even with the availability of IFN data (which may peak earlier than 5 dpi). Thus, it is necessary for the authors to perform a sensitivity analysis regarding the fixed parameters.

Reviewer #2: The authors addressed all my comment.

**Have the authors made all data and (if applicable) computational code underlying the findings in their manuscript fully available?**

Reviewer #1: Yes

Reviewer #2: Yes

PLOS authors have the option to publish the peer review history of their article (what does this mean?). If published, this will include your full peer review and any attached files.

Reviewer #1: No

Reviewer #2: No

Figure Files:

Data Requirements:

Reproducibility:

References:

---

## [Decision Letter · Decision Letter 2]

12 Jul 2023

Dear Student MARC,

We are pleased to inform you that your manuscript 'Impact of variants of concern on SARS-CoV-2 viral dynamics in non-human primates.' has been provisionally accepted for publication in PLOS Computational Biology.

Best regards,

Rustom Antia

Academic Editor

PLOS Computational Biology

Amber Smith

Section Editor

PLOS Computational Biology

Reviewer's Responses to Questions

**Comments to the Authors:**

Reviewer #1: Now they answer to all of my comments, congratulations!

**Have the authors made all data and (if applicable) computational code underlying the findings in their manuscript fully available?**

Reviewer #1: None

PLOS authors have the option to publish the peer review history of their article (what does this mean?). If published, this will include your full peer review and any attached files.

Reviewer #1: No

---

## [Editor Report · Acceptance letter]

3 Aug 2023

PCOMPBIOL-D-22-01635R2 

 Impact of variants of concern on SARS-CoV-2 viral dynamics in non-human primates. 

Dear Dr MARC,

I am pleased to inform you that your manuscript has been formally accepted for publication in PLOS Computational Biology. Your manuscript is now with our production department and you will be notified of the publication date in due course.

With kind regards,

Zsofi Zombor
